

# The seismic hazard from the Lembang Fault, Indonesia, derived from InSAR and GNSS data

Ekbal Hussain[1], Endra Gunawan[2], Nuraini Rahma Hanifa[3], and Qori'atu Zahro[3]

[1]British Geological Survey, Natural Environment Research Council, Keyworth, Nottingham, NG12 5GG, UK
[2]Global Geophysics Research Group, Faculty of Mining and Petroleum Engineering, Bandung Institute of Technology, Bandung 40132, Indonesia
[3]Research Center for Geological Disaster, National Research and Innovation Agency, Bandung 40135, Indonesia
**Correspondence:** Ekbal Hussain (ekhuss@bgs.ac.uk)

**Abstract.** A growing number of large cities are located near poorly understood faults that have not generated a significant earthquake in recent history. The Lembang Fault is one such fault located near the city of Bandung in West Java, Indonesia. The slip rate on this fault is debated with estimates ranging from 6 mm/yr to 1.95–3.45 mm/yr, derived from GNSS campaign and geological measurements respectively. In this paper we measure the surface deformation across the Bandung region and
resolve the slip rate across the Lembang Fault using radar interferometry (InSAR) analysis of 6 years of Sentinel-1 satellite data and continuous GNSS measurements across the fault. Our slip rate estimate for the fault is 4.7 mm/yr, with the shallow portions of the fault creeping at 2.2 mm/yr. Previous studies have estimated the return period of large earthquakes on the fault to be between 170–670 years. Assuming simplified fault geometries and a reasonable estimate of the seismogenic depth we derive an estimated moment deficit of a magnitude 6.8–7.2 earthquakes; indicating that the fault poses a very real risk
to the local population. Using the Global Earthquake Model OpenQuake-engine we calculate ground motions for these two earthquake scenarios and estimate that 2.5–3.3 million people within the Bandung Metropolitan region would be exposed to ground shaking greater than 0.3g. This study highlights the importance of identifying active faults and understanding their past activity, and measuring the current strain rates of smaller crustal active faults located near large cities in seismic zones.

## 1 Introduction

More than half of all people in the world now live in increasingly dense urban centres (Ritchie and Roser, 2018). This transition to a more urbanised world has enabled the social and economic development of millions of people, particularly in low-middle income nations. However, many of these cities are located near active faults that have not generated a significant earthquake in recent memory, raising the risk of losing hard-earned progress through a devastating future earthquake (Amey et al., 2021, 2022; Elliott, 2020; Hussain et al., 2020).

With a population of 8.4 million (2021 estimates) Bandung, the capital of West Java, Indonesia, is one such city. The centre of the city lies less than 10 km south of the Lembang Fault (Figure 1), a major crustal fault in west Java. Although there are no documented records of large historical earthquakes, the Lembang Fault shows geomorphic evidence of recent activity (Daryono et al., 2019).



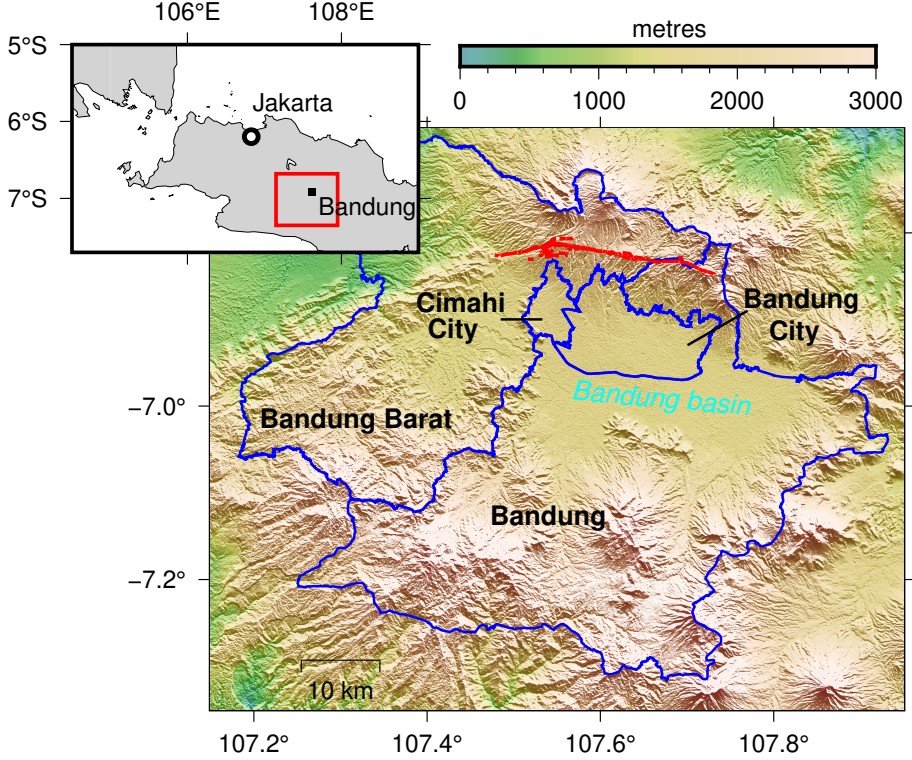

**Figure 1.** The Bandung Metropolitan region in west Java, Indonesia. The majority of the population in the region live in Bandung and Cimahi cities, which sits in the Bandung basin. The basin is bounded by mountains on all four sides with the Lembang Fault (in red) to the north.

The fault dips to the north by about 75 degrees and is a prominent landmark of slope breaks between a series of east-west-trending linear ridges that separate the north Bandung highland from the wide and flat Lembang basin to the north (Daryono et al., 2019). Large-scale geomorphic mapping and dating of offset volcanic material indicate that the fault is separated into two parts; an older eastern section and the younger western part (Dam et al., 1996). The eastern Lembang Fault is thought to have originally formed from a large-scale sector collapse of the Sunda volcano 126-135 ka, due to the rapid depressurisation of the magma chamber (Van Bemmelen, 1949; Dam et al., 1996), while the western Lembang Fault is thought to be active since around 24 ka (Dam et al., 1996). Fault segmentation is also seen in the topographic expression of the fault because the eastern part has a much larger topographic relief across the fault (200-400m), while the western part has a smaller offset (10-150m) but is still steep and prominent.

While there have been no records of major events on the fault, small earthquakes have been detected in recent times with near-field instruments. Moment tensors from two magnitude 3.4 earthquakes in 2011 and magnitude 2.8 and 2.9 events in 2017 reveal left-lateral motion across the fault (Sulaeman and Hidayati, 2011; Sanny, 2017; Daryono et al., 2019; Nugraha et al., 2019). Daryono et al. (2019) mapped the surface expression of the fault to determine its length at 29 km long. During trenching investigations they found evidence of at least 3 large earthquakes in the 15th century, 2300–60 BCE and 19,620–19,140 BP,





with the displacement across the 2300–60 BCE event equivalent to a Mw 6.5 earthquake. By measuring and dating geomorphic offsets of river basins across the fault they estimate a long term left-lateral fault slip rate of 1.95–3.45 mm/yr. Overall they

estimate that the fault could produce a Mw 6.5–7 earthquake with a recurrence time of 170–670 years.

However Meilano et al. (2012), using GPS measurements between 2006 and 2011 to determine a geodetic left-lateral fault slip rate equivalent to 6 mm/yr, nearly twice that determined by geological measurements. Meilano et al. (2012) also estimate shallow creep on the fault at the same rate, implying that the shallow portion of the fault (<3 km) is not accumulating any seismic strain.

In this paper we will measure the surface deformation and resolve the slip rate across the Lembang Fault using radar interferometry (InSAR) analysis of freely available radar data from the European Space Agency's Sentinel-1 satellite constellation and unpublished continuous GPS data in the region. We will resolve the slip rate and creep rate of the fault. Using this rate we will estimate the potential size of an earthquake given a 600 year return period and calculate the potential shaking to the Bandung Metropolitan area.

## 2    InSAR data analysis


For the InSAR processing we used Sentinel-1 Single Look Complex images over the Bandung metropolitan region (Longitude, Latitude: 107.62, -6.93). In total we downloaded 146 images from ascending track 98 spanning 4 January 2015 and 27 December 2020, and 154 images from descending track 149 spanning 7 January 2015 and 30 December 2020. We performed initial InSAR processing using the Interferometric synthetic aperture radar Scientific Computing Environment (ISCE) software

(Rosen et al., 2012; ISCE2, 2021), which resulted in 432 and 450 small baseline interferograms respectively for ascending track 98 and descending track 149. To improve the signal-to-noise ratio we multi-looked each interferogram by 2 pixels in azimuth and 6 pixels in range giving pixel sizes of approximately 30 m.

For the time series analysis, we processed each interferogram stack independently using the Miami INsar Time-series software in PYthon (MintPy) (Yunjun et al., 2019, 2020). The small baseline interferogram networks are shown in Figure 2. We

removed interferograms with an average spatial coherence less than 0.35. This resulted in 18 and 15 interferograms being removed from the ascending and descending tracks respectively, which are represented by dashed lines in Figure 2. We used the SRTM 30m digital elevation model (Farr et al., 2007) to estimate and correct the topographic error. We also estimated and removed the atmospheric contribution to the phase in each interferogram using the GACOS weather model reanalysis product (Yu et al., 2018). Additionally, we removed a bilinear ramp from each interferogram to reduce any remaining long wavelength

trends. We detect and correct unwrapping errors in the interferograms using the phase closure functionality in MintPy (Yunjun et al., 2019).

The average line-of-sight (LOS) velocities for the descending and ascending tracks are shown in Figure 3(a, b), where blue colours represent motion away from the satellite. The black square is the spatial reference. Using the method described by Wright et al. (2004) we exploit the different viewing geometries of the same location to deconvolve the ascending and

descending velocities into east-west and vertical. This requires the assumption that the relative north motion is negligible,




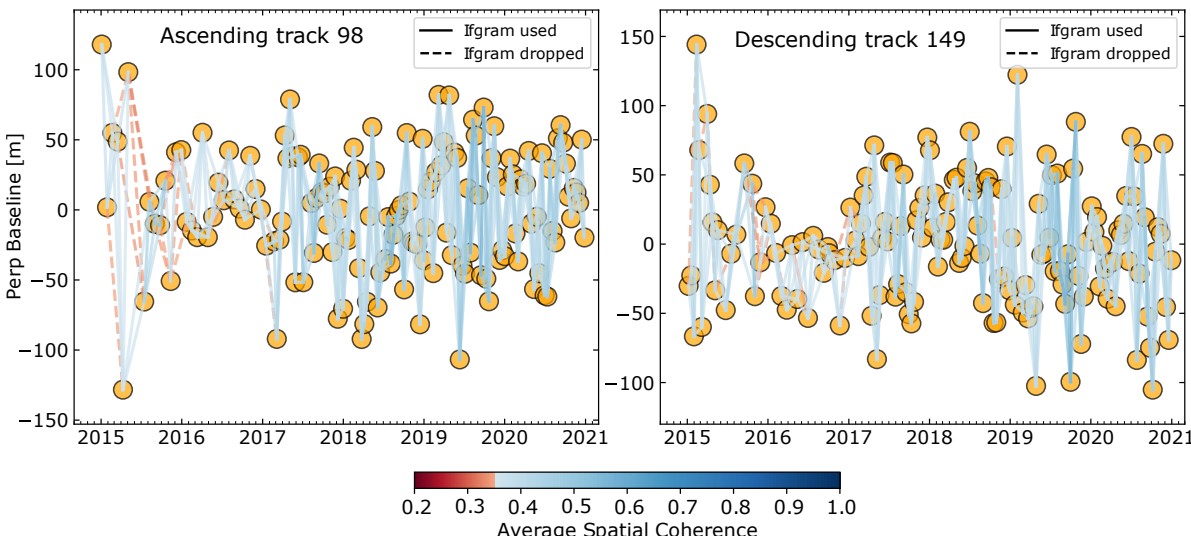

**Figure 2.** The small baseline interferogram network. The yellow circles represent the satellite acquisition with the lines representing the interferograms colour coded by the average spatial coherence. The dashed lines are interferograms dropped due to poor coherence.

which is a common assumption in InSAR analysis as polar orbiting Sentinel-1 satellites are less sensitive to motion in that direction. The deconvolved velocities are given in Figure 3(c, d).

While the focus of this paper is the horizontal velocities (Figure 3d), it is worth noting that the deconvolved velocities also show a pattern of vertical deformation within the Bandung basin (Figure 3c). This is generally thought to be subsidence due to groundwater extraction (e.g. Abidin et al., 2008; Ge et al., 2014; Du et al., 2018).

## 3 GNSS data analysis

A continuous GNSS network, consisting of three continuous stations, was installed on 7 September 2019 using dual frequency Trimble GPS receivers. These GNSS stations, named LMB1 in the south, LMB2 in the central, and LMB3 to the north of the Lembang fault, are an addition to the existing GNSS stations of the Indonesian Geospatial Information Agency, CLBG (Gunawan and Widiyantoro, 2019; Gunawan, 2021). In concern to the stability of the GNSS network, we installed the GNSS antenna on top of a concrete building. We use electricity to power up the receiver, which is located inside a storage box. In total, this study used four GNSS stations available perpendicular to the Lembang fault (Figure 4).

In this network, GNSS data was recorded with a 30 seconds sampling interval. The 30s RINEX data were processed using the GipsyX software, which was developed by Jet Propulsion Laboratory (JPL) (Bertiger et al., 2020). We conduct static estimation using precise point positioning implemented in the software by selecting fiducial-free with five iterations. Additional files, such as final orbits, clocks and other products, are from the Jet Propulsion Laboratory. All of those products are based on the JPL's reanalysis final set of the International GNSS Service IGS14 of the ITRF14 reference frame (Altamimi et al., 2016). We





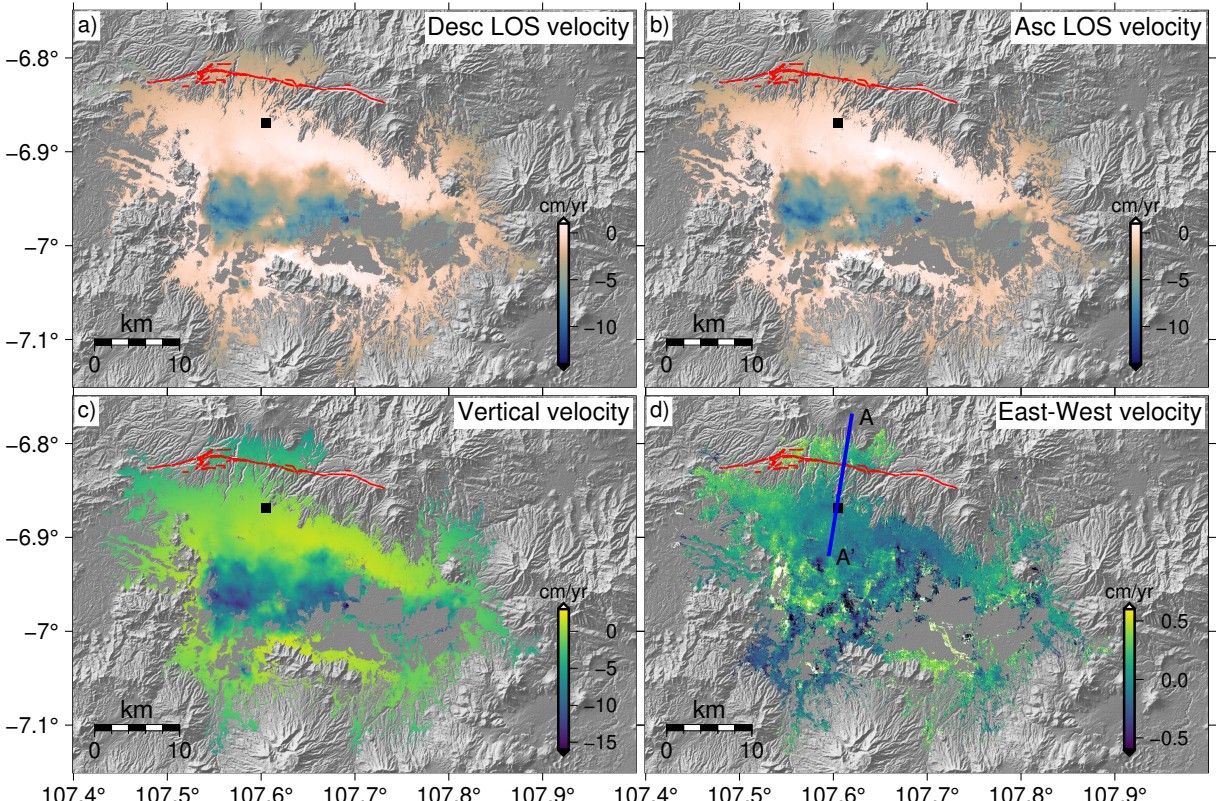

**Figure 3.** The average descending and ascending line-of-sight velocities (a, b respectively). Blue colours represent motion away from the satellite. The black square is the spatial reference for each map. c) and d) are the deconvolved vertical and east-west components of ground motion, assuming the relative north component is negligible. Blue colours represent subsidence in the vertical, and motion to the west in the east-west velocities. The red lines are the Lembang fault segments mapped by Daryono et al. (2019).

also used the GOT4.8 model to correct for ocean tidal loading, which is calculated from the website of the Onsala Space Observatory (http://holt.oso.chalmers.se/loading). In each of our RINEX data, we set an elevation angle cutoff with 15°. Using the time series data obtained from the analysis, we computed the velocity for each GNSS station. The velocity for each GNSS station at mm level. Horizontal velocities for each GNSS station are shown in Figure 4.

## 4 Slip rate estimates

To investigate the change in velocity, and therefore the strain accumulation, across the Lembang fault we projected the east-west velocities across the fault onto a single fault perpendicular profile (A-A' in Figure 3d), including all points 5 km either side of the profile. The results (grey points in Figure 5) reveal a change in velocity across the Lembang fault consistent with left-lateral motion.





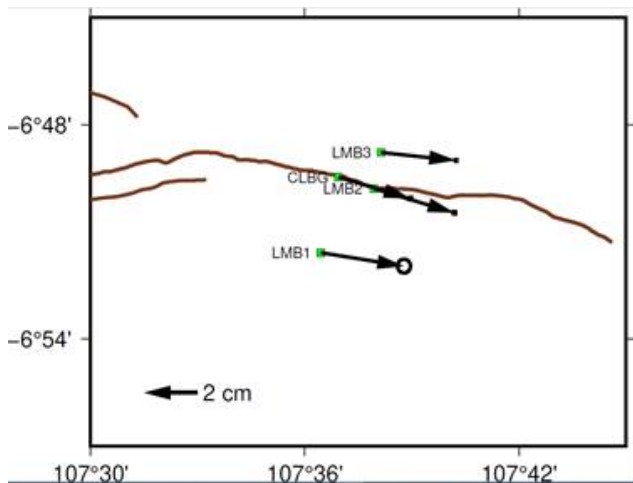

**Figure 4.** Horizontal velocities at GNSS stations in the ITRF2014 reference frame. Brown lines marks the delineation of the Lembang fault.

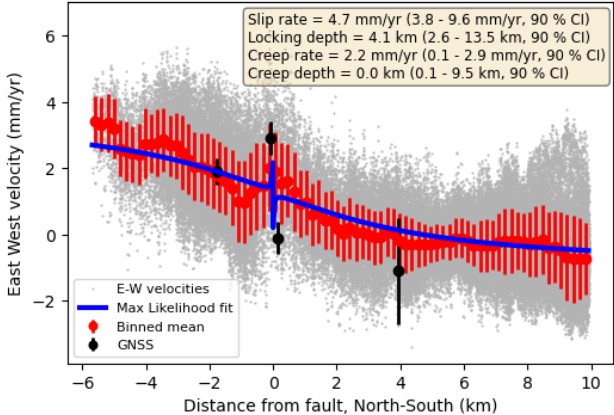

**Figure 5.** The east-west component of the InSAR (grey) and GNSS velocities (black) across the Lembang Fault along profile A-A' in Figure 3d. Positive velocities are motion to the west. 3d. The red points are the mean and one standard deviation of the InSAR velocities in 200 m bins along the profile. The maximum likelihood solution of the MCMC model is shown in blue with the best fit parameters in the text box. The 90% confidence interval (CI) for each paremeter is given in the parenthesis.

To better reflect this velocity and the uncertainty in the data we binned the values in 200 m windows along the profile. These are shown in red with error bars representing the one-sigma standard deviation. The GNSS velocities are also shown in black. To estimate the fault parallel slip rate ($S$) and creep rate ($C$) from the velocity profile ($v_x$) we fit the 1-D screw dislocation

model of Hussain et al. (2016), adapted from Savage and Burford (1973):

$$v(x) = -\frac{S}{\pi} arctan\left(\frac{x}{d_1}\right) + C\left[\frac{1}{\pi} arctan\left(\frac{x}{d_2}\right) - \mathcal{H}(x)\right] + a$$



Where $x$ is the perpendicular distance from the fault, $d_1$ is the interseismic locking depth, $d_2$ the creep depth, $\mathcal{H}(x)$ is the Heaviside function and $a$ is a static offset.

We find best fit values for each model parameter ($S$, $d_1$, $C$, $d_2$, and $a$), using the emcee implementation of the Goodman and Weare (2010) affine-invariant ensemble Markov Chain Monte Carlo (MCMC) sampler (Foreman-Mackey et al., 2013).

Our MCMC sampler explores the parameter space constrained by $0 < S$ (mm/yr) $< 10$, $0 < d_1$ (km) $< 20$, $0 < C$ (mm/yr) $< 10$, $0 < d_2$ (km) $< 10$, -10 $< a$ (mm/yr) $< 10$, assuming a uniform prior distribution over each range. We run the model for 500,000 steps, allowing 300 for burn, and thin the results in steps of 20. This produces 248,500 random samples from which we estimate both the maximum likelihood solution for each parameter and its uncertainties.

The maximum likelihood solution is shown by the blue line in Figure 5, which corresponds to a slip rate of 4.7 mm/yr, a
locking depth of 4.1 km, and a shallow surface creep rate of 2.2 mm/yr.

The full distribution of MCMC samples are shown in Figure 6 with the maximum likelihood estimate for each parameter shown by the blue square. The sampled marginal probability distribution for the slip rate has a constrained lower bound with an unconstrained upper tail while the distributions for the locking depth and creep rate are approximately normal. The creep depth has a peak around zero with a poorly constrained upper bound. As is commonly seen in elastic dislocation models there
is a trade-off between the fault slip rate and the locking depth (e.g. Wright et al., 2013; Hussain et al., 2016) where a slower slip rate can be compensated by a shallower locking depth.

## 5   Scenario earthquake hazard

The Lembang fault has not recorded a large earthquake (>Mw 5) during the modern instrumental period. However there is geomorphological evidence of significant size earthquakes in the Quaternary period. Based on measurement of offset rivers
and palaeoseismic trenching Daryono et al. (2019) estimate that the fault could produce a large magnitude earthquake with a recurrence time of 170–670 years.

We can now use the magnitude scaling relationship of Hanks and Kanamori (1979): $Mw = \frac{2}{3}log_{10}(M_0) - 10.7$ with $M_0 = \mu AST$, where $\mu$ is the rigidity, $A$ the fault area, $S$ the slip rate and $T$ the return period, to estimate the potential size of earthquakes on the fault. Assuming our estimated fault slip rate of 4.7 mm/yr remains constant over a time, a fault length of
29 km (Daryono et al., 2019), constant fault dip of $60°$, and a seismogenic depth of 15 km based on microseismicity relocation depths (Supendi et al., 2018), gives an estimated moment deficit of a magnitude 6.8–7.2 earthquake for return periods between 170–670 years; indicating that the fault poses a very real risk to the local population.

To explore the potential exposed population in the Bandung Metropolitan region to high levels of ground shaking resulting from an earthquake on the Lembang Fault we used the GEM OpenQuake-engine v3.14 (Pagani et al., 2014; GEM, 2022) to
calculate the ground motion fields for two potential Lembang Fault scenarios: a magnitude 6.8 and a magnitude 7.2 earthquake. We modelled the Lembang Fault as two rectangular slip planes (total length of 29 km) to account for the changes in geometry along strike (brown line in Figure 7). Both planes dip at $60°$ to the south (Daryono et al., 2019) from the surface down to 15 km.





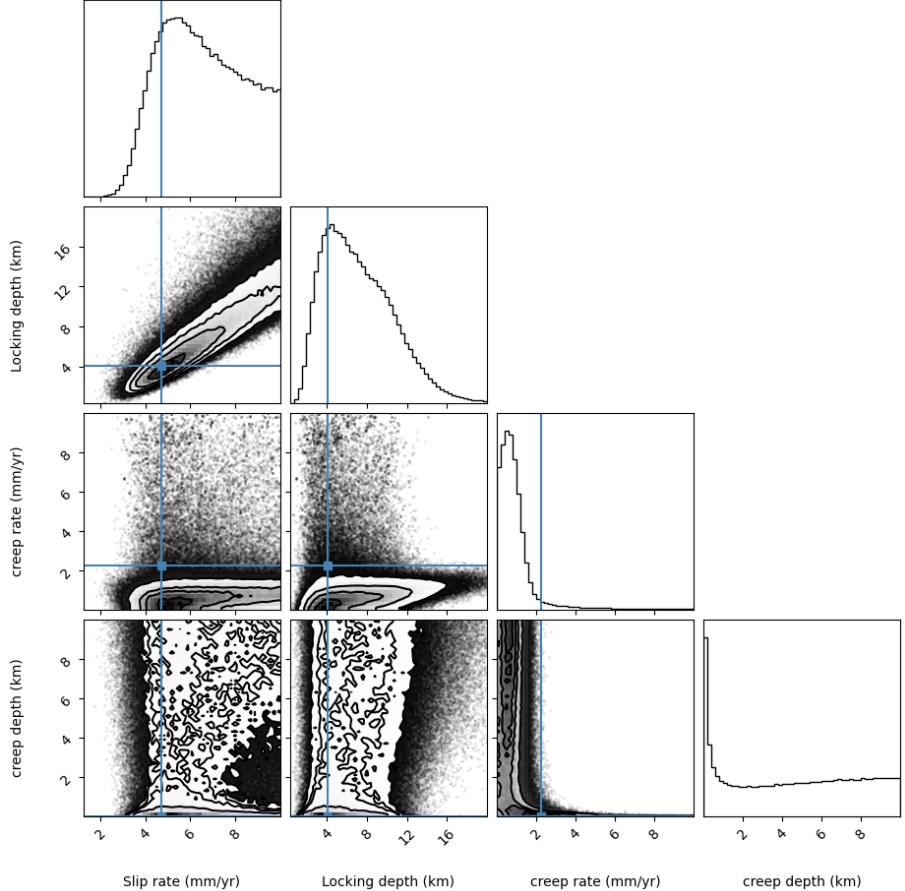

**Figure 6.** The full distribution of MCMC samples. The blue square marks the maximum likelihood estimate, which corresponds to a fault slip rate of 4.7 mm/yr, a locking depth of 4.1 km, and a shallow creep rate of 2.2 mm/yr, and a creep locking depth of 0 km.

The hazard calculation involves determining the spatial pattern of Peak Ground Acceleration (PGA) for each scenario event

by using a Ground Motion Prediction Equation (GMPE). There are many GMPEs available in the literature (see Douglas and Edwards (2016) for a review and www.gmpe.org.uk for an updated compendium managed by Dr John Douglas). In our model we used three equally weighted equations for active shallow crustal earthquakes: Abrahamson et al. (2014); Campbell and Bozorgnia (2014); Chiou and Youngs (2014). Averaging several GMPEs helps to partially propagate the epistemic uncertainty of the distribution of shaking arising from an imperfect knowledge of ground motion.

For each magnitude scenario we assume the entire fault ruptures in the earthquake and produce 1000 realisations of the ground motion to account for aleatory variability. We also account for the spatial correlation of ground shaking during the generation of each ground-motion field, to ensure assets located close to each other will have similar ground-motion levels, according to the methods described by Jayaram and Baker (2009).



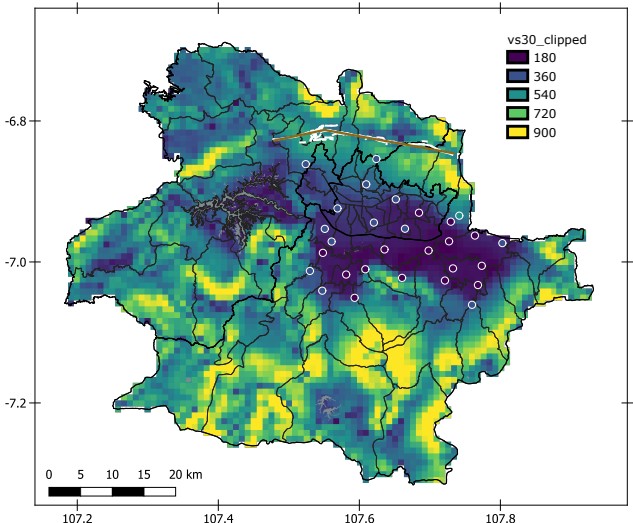

**Figure 7.** Vs30 velocities - the shear wave velocity in the top 30 m of soil for the Bandung Metropolitan Region from the USGS Global Vs30 Map Server (Wald and Allen, 2007; Allen and Wald, 2009). Velocities are estimated using topographic slope as a proxy, assuming stiffer materials with faster Vs30 values are more likely to maintain a steep slope and flat sedimentary basins are characterised by a lower velocity. The white bordered circles are the point measurements of Vs30 velocities from a microtremor survey by Pramatadie et al. (2017). The white lines are the mapped Lembang fault trace by Daryono et al. (2019). The brown lines are the simplified fault model used in our seismic hazard calculations.

The majority of the population of the Bandung Metropolitan Region live in the Bandung basin, which is a large 2,300 km$^2$
sedimentary basin bounded on all 4 sides by mountains (Figure 1 and Figure 9). Deposits in the basin comprise of coarse volcaniclastics, fluvial sediments and notably a thick series of lacustrine deposits (Van der Kaars and Dam, 1995).

Deep sedimentary basins increase the amplitude and duration of earthquake ground motions from seismic waves, increasing the seismic hazard for cities sited on such basins (e.g. Bard and Bouchon, 1985; Bard et al., 1988; Rial, 1989; Bielak et al., 1999; Wirth et al., 2019). We attempt to account for basin amplification in our ground motion model by using the Vs30 velocities
- the shear wave velocity in the top 30 m of soil. Pramatadie et al. (2017) conducted a microtremor survey of the Bandung basin and calculated the shallow shear wave velocity for 29 sites across the basin. We supplemented these point measurements with estimated Vs30 velocities from the USGS Global Vs30 Map Server (Wald and Allen, 2007; Allen and Wald, 2009). This method derives maps of seismic site conditions using topographic slope as a proxy. This assumes that maintaining a steep topographic slope requires stiffer materials (higher Vs30 values), while deep basin sediments are deposited mainly in
environments characterised by a lower velocity. Note that we recognise that this will probably be an underestimate of the full basin effects, which would require detailed models of the 3D geometry and sediment type to fully account for (Joyner, 2000). Figure 7 shows the Pramatadie et al. (2017) point measurements of Vs30 velocities match the USGS Global Vs30 estimates very well.



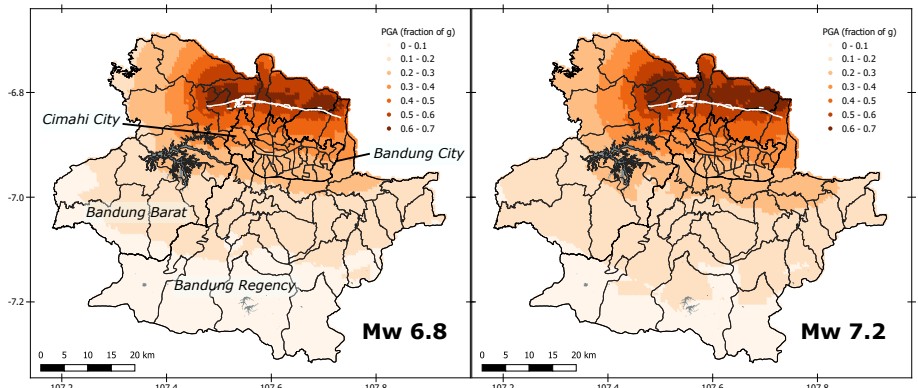

**Figure 8.** Median ground motion fields for a moment magnitude 6.8 (left) and a magnitude 7.2 (right) earthquake on the Lembang Fault (white lines). Peak Ground Accelerations (PGA) are shown as a fraction of *g*. As expected, the highest shaking is focused to the north of the Lembang fault over the hanging wall.

The estimated ground motions for the two earthquake scenarios are shown in Figure 8 where darker colours represent higher expected Peak Ground Acceleration (PGA), given as a fraction of *g*. The maps show high accelerations to the north of the fault, which is expected because the fault dips to the north meaning points north of the fault are closer to the rupture plane and therefore likely to experience faster accelerations than points to the south.

### 5.1 Exposed population

The Bandung Metropolitan region comprises of 4 administrative areas with a high variation in population. Population secondary data is taken from the Central Bureau of Statistics and the Civil Registry Office, where the population for each district is updated regularly. However, during the COVID-19 pandemic some regions did not update their population, in this case the number of residents was calculated by projection and estimation.

Population projection prepared using the Linear Arithmetic Growth Model, assume that population in the future will increase linearly with the following formula:

$$P_t = P_0(1 + rt)$$

where, $P_t$ is the current year population, $P_0$ is the base year population, $r$ the growth rate, and $t$ the time interval.

By overlapping the predicted ground motion fields shown in Figure 8 with the population map we find that 2.5 million people are potentially exposed to high levels of ground shaking (>0.3g) in the event of a Mw 6.8 earthquake on the Lembang Fault. This increases to 3.3 million people for a magnitude 7.2 earthquake.

Note that such simple overlaps are insufficient to establish the true risk to earthquake shaking. Most fatalities in non-coastal earthquakes occur due to building collapse (Coburn et al., 1992; Ambraseys and Bilham, 2011). Therefore a thorough assessment of the fragility of buildings when exposed to certain levels of ground shaking will be needed to assess the risk. Since this would need to be done for all possible building types in the metro region it is beyond the scope of this study.





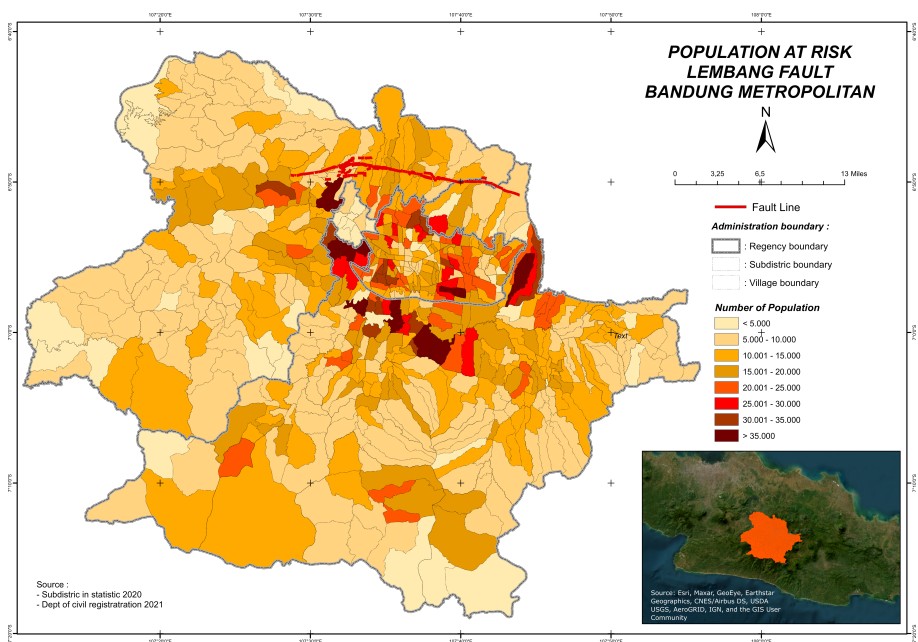

**Figure 9.** The population distribution in the Bandung Metropolitan Region. Data source: Subdistrict in statistio 2020, Department of Civil Registration2021. The red line is the Lembang Fault trace.

## 6 Discussion

### 6.1 Lembang fault parameters

The maximum likelihood estimate for the geodetic slip rate on the Lembang Fault from our analysis using a combination of InSAR and GNSS data is 4.7 mm/yr, which falls between the 1.95–3.45 mm/yr estimated by Daryono et al. (2019) using geomorphic offsets and paleoseismic trenching, and the 6 mm/yr estimated by Meilano et al. (2012) using GNSS measurements. While the 90% confidence interval for our slip rate (3.8–9.6 mm/yr) encompasses the estimate by Meilano et al. (2012), it is still higher than even the upper bound of the geological measurements by Daryono et al. (2019). This discrepancy between the geological and geodetic slip rate could be due to the uncertainties associated with the measurement and dating of geomorphic offsets. Fluvial channels and other geomorphic features in tectonically active regions constantly change, as degradation can begin immediately after formation. The longer-lived and larger the offset feature, the greater the uncertainty becomes, making slip estimates more difficult to interpret (Arrowsmith et al., 2012; Scharer et al., 2014).

Our maximum likelihood estimate for the locking depth is 4.1 km. This is almost certainly an underestimate of the true locking depth on the fault. However, the 90% confidence bound on the locking depth is: 2.6–13.5 km showing that there is significant uncertainty on our estimate. The upper bounds of this range is closer to the seismologic depth for this region (Supendi et al., 2018). Additionally, small earthquakes on or around the fault show that there is strain accumulation and release




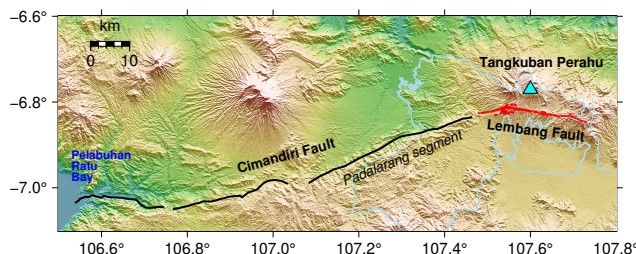

**Figure 10.** The Cimandiri Fault is a ∼100 km long multi-segment fault located to the west of the Lembang Fault (Marliyani et al., 2016).

at depth. For example, a magnitude 3.4 earthquake in July 2011 had a focal depth of ∼6 km (Sulaeman and Hidayati, 2011), and magnitude 2.8 and 2.9 earthquakes in 2017 occurred at depths of ∼5 km (Nugraha et al., 2019).

Meilano et al. (2012) in their study suggested that the Lembang fault is creeping at shallow depths (down to 3 km) at 6 mm/yr, at the same rate as the fault slip rate. If this is true then it means that there is no tectonic strain accumulation on the
shallow portions of the fault. However our maximum likelihood creep rate on the fault is 2.2 mm/yr, which is significantly less than Meilano et al. (2012)'s estimate, and less than half the full fault slip rate (4.7 mm/yr). Meaning that there is still strain accumulation even on the creeping segment of the fault. It is unclear how Meilano et al. (2012) determined the depth extent of their creep estimate. It is possible that the 3 km depth was a prescribed number in their model. Figure 6 shows a significant uncertainty on the creep depth with the upper bound completely unconstrained by the model. However, the peak corresponds
to 0 km, implying that the creep is most likely confined to shallow depths.

## 6.2 Multi-segment ruptures with the Cimandiri Fault zone

The Cimandiri Fault zone is a ∼100 km long ENE striking fault located to the west of the Lembang Fault and continuing to the Pelabuhan Ratu Bay (Marliyani et al., 2016) (Figure 10). The fault zone is expressed in the bedrock by numerous NE, west, and NW trending thrust and strike-slip faults and folds. The predominant sense of motion is left-lateral on the eastern segments
with more thrust dominant motion at the western end (Marliyani et al., 2016; Supendi et al., 2018).

The separation between the surface trace of the western end of the Lembang Fault and the eastern end of the Cimandiri Fault is less than ∼5 km. While some authors suggest that the two faults are not connected at depth (Mahya and Sanny, 2017), most agree that the broader strain pattern is consistent across the two faults (e.g. Abidin et al., 2009; Kopp, 2011; Marliyani et al., 2016; Daryono et al., 2019). Marliyani et al. (2016) used dislocation and geometric models to show that the segmentation of
the Cimandiri fault zone could act as a barrier to earthquakes along the zone, thereby limiting the overall size of potential earthquakes. Nevertheless, large earthquakes have been known to jump larger across larger segment distances, most notably 50 km in the 1997 Pakistan earthquake (Nissen et al., 2016).

Additionally an earthquake on the Lembang Fault or the eastern portion of the Cimandiri fault (the Padalarang segment (Figure 10)) could change the stress state of the neighbouring fault thereby raising the possibility of a triggered event or
progressive failure of the adjacent fault, particularly if the faults are late in its earthquake cycle and critically stressed (e.g.





Stein et al., 1997; Kilb et al., 2002; Nissen et al., 2016; Mildon et al., 2019). There are no records of historical earthquakes along the Padalarang segment with a magnitude greater than 6; but with continuing strain accumulation (Marliyani et al., 2016), a triggered event remains a distinct possibility, and therefore an additional source of potential seismic hazard to the Bandung Metropolitan Region.

Based on simple length-width scaling relationships (Stirling et al., 2008), Marliyani et al. (2016) estimate that the Padalarang segment of the Cimandiri Fault could be rseponsible for a magnitude 6.9 earthquake. Based on our analysis of the strain accumulation along the Lembang Fault, a combined Padalarang-Lembang rupture could therefore result in a potential magnitude 7.3 earthquake (assuming a 670 year return period for the Lembang Fault).

### 6.3   Caveats and Limitations

A major source of uncertainty in seismic hazard calculations is in the choice of the Ground Motion Prediction Equations (GMPEs). To partly mitigate this effect and in order to capture some of the epistemic uncertainty in median ground motion estimates we used several equally weighted GMPEs tailored to shallow crustal-tectonic settings (Abrahamson et al., 2014; Campbell and Bozorgnia, 2014; Chiou and Youngs, 2014). Additionally, to capture the aleatory variability within each GMPE we allow for a 3-sigma variability within each model. Nevertheless there remains significant uncertainty in these calculations.

The aim of this paper has been to investigate the directly exposed populations to high levels of ground shaking from potential earthquakes by overlapping the population count with the predicted ground shaking. However, this simplifies the true exposure of people who at any one point could be inside a building of variable type and therefore fragility, out on the streets, in open spaces such as parks and therefore less vulnerable to injury. Additionally the timing of the earthquake also matters. An earthquake at night will mean more people are exposed to shaking in their homes while in the day the exposure is highest at

places of work or in transit (Freire and Aubrecht, 2012).

It is also important to remember the geographical and geological context of the study area. The majority of the population in the Banding Metropolitan Region live in the Bandung basin, which is a sedimentary basin bounded by volcanic highlands. A large earthquake on the Lembang Fault could result in multiple secondary hazards such as liquefaction, landslides, and blocked waterways leading to floods, fires, etc. Additionally, recent studies have shown that large tectonic earthquakes are capable

of triggering volcanic eruptions (e.g. Bedón et al., 2022; Jenkins et al., 2022; Sinaga et al., 2022). Meaning that a Lembang Fault earthquake could potentially trigger an eruption at the Tangkuban Perahu stratovolcano located 30 km north of Bandung (Figure 10). Such hazard cascades can lead to dynamic changes to exposure and vulnerability and potentially increased losses in terms of lives and livelihoods (Gill and Malamud, 2014; de Ruiter and Van Loon, 2022).

### 7   Conclusions

In this study we used 6 years of Sentinel-1 Sentinel-1 radar data to produce deformation maps for the Bandung metropolitan region. By combining the ascending and descending velocities we calculate the east-west velocities across the region and show that there is a strain concentration on the Lembang Fault. We model the velocity profile across the fault using a simple screw



dislocation model allowing for shallow creep and find that the slip rate across the fault is 4.7 mm/yr with the shallow portions of the fault creeping at 2.2 mm/yr, less than half the full slip rate implying that strain is still accumulating at shallow depths.

Assuming reasonable fault geometries and estimated return periods of 170–670 years we derive an estimated moment deficit on the fault equivalent to earthquakes of magnitude 6.7–7.2. We estimate the expected ground shaking from these two earthquakes using the Global Earthquake Model OpenQuake-engine, taking into the effect of basin amplification using a combination of measured and estimated Vs30 velocities. Our results show that 2.5–3.3 million people within the Bandung Metropolitan region would be exposed to high levels of ground shaking (greater than 0.3g) given these earthquake scenarios respectively.

In subduction zone settings the megathrust is often the main focus of research for seismic hazard analysis. Our work here highlights the importance of not forgetting local crustal faults located near large urban centres, which also pose a high hazard to communities.

*Code and data availability.* The Sentinel-1 SAR data is avaialble for free from the Copernicus open access web portal (https://scihub. copernicus.eu). The ISCE2 InSAR processing software is available on github (https://github.com/isce-framework/isce2). The MintPy In-

SAR time series processor is available on github (https://github.com/insarlab/MintPy). The Global Earthquake Model OpenQuake engine can be downloaded from github (https://github.com/gem/oq-engine).

*Author contributions.* EH and NRH conceived the research framework and developed the methodology. EH was responsible for the InSAR data analysis, and graphic visualisation. EG was responsible for the GNSS data collection and processing. QZ was responsible for the population exposure estimates. EH, NRH and EG wrote the first draft. All the authors discussed the results and contributed to the final

version of the paper.

*Competing interests.* The authors declare no competing interests.

*Acknowledgements.* The authors would like to thank Professor Gavin Sullivan for initiating the Indonesia Seismic Cities project that enabled this collaboration. This research was funded by the BGS-NERC ODA grant NE/R000069/1 'Geoscience for Sustainable Futures', and the BGS International National Capability programme 'Geoscience to tackle Global Environmental Challenges', NERC reference

NE/X006255/1. The Indonesian counterparts were funded by Rumah Program Kebencanaan 2022 research grant managed by The National Research and Innovation Agency of Indonesia (BRIN)'s Research Organization for Earth Sciences and Maritime, the National Competitive Basic Research (PDKN) Program of the Ministry of Education, Culture, Research, and Technology and the Overseas Research Grants of Asahi Glass Foundation. The paper is published by permission of the Director of the British Geological Survey.



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
