# Peer review of "The seismic hazard from the Lembang Fault, Indonesia, derived from InSAR and GNSS data"

_EGUsphere, 2022_

## Referee Comment (RC2)

**Review of**
**The seismic hazard from the Lembang Fault, Indonesia,**
**derived from InSAR and GNSS data**
**by E. Hussain et al. (2023)**

In this study, the authors estimate the slip and shallow creep rates of the Lembang fault, Indonesia, by means of radar interferometry and continuous GNSS time series analysis. The results are used to make a first order assessment of the seismic hazard posed by this fault to the population living in neighbouring areas. This assessment is based on the superposition of the size of the population distribution and the ground accelerations predicted by the Global Earthquake Model OpenQuake engine for two maximum magnitude earthquakes (full rupture of the fault), corresponding to the moment deficit accumulation derived from the resulting slip rate and the upper and lower limits of the return period of large earthquakes on the fault.

This is, thus, a study of a poorly understood fault which might have been overlooked because it hasn't had any recorded historical seismic activity. However, previous paleoseismology studies indicate large crustal earthquakes have been produced at this fault, with return periods longer than a century, and the present work confirms strain accumulation is occurring at the fault at rates that make it capable of producing shallow earthquakes with magnitudes larger than 6.5, at a fault embedded in a densely populated area. These are important results, not only to better understand the fault behaviour and to contribute to hazard assessment, but also to inform urban planning and risk for the nearby populations.

The manuscript is very well written and has very nice figures that illustrate and support the main text. It is my opinion that the manuscript is almost ready for publication. However, I noted a few places where the discussion could be refined in order to improve its clarity. I list these points in the comments below.

**Main comments:**

1. In the beginning of the manuscript, the authors state that "The fault dips to the north by about 75 degrees" (line 24). Why is a "constant fault dip of 60°" (line 125) used to estimate the area of the fault and the ground motion fields (line 132)? Also, when estimating the ground motion fields, the fault is modeled as two fault planes that "dip at 60° *to the south*" (line 132). Why the different orientation?
2. In the manuscript it is assumed that the motion at the fault plane is exclusively left-lateral strike-slip. Is there no evidence at all of dip-slip motion? Do the SAR and GNSS observations confirm this?
3. In figures 3a, b, c, the strong signal attributed to subsidence due to groundwater extraction obscures the signal due to interseismic deformation, which is the target of this study. Perhaps a change in the limits of the color scale could help enhance (at least in Figure 3c) the velocity gradient at the fault (if any), in order to confirm the assumption of no dip-slip rate accumulation.

This could be done on a separate panel or an inset showing only the area of interest. A figure with profile A-A' showing a negligible vertical gradient could also help.

4. Figure 3d could also benefit from a change in the color scale to enhance the horizontal velocity gradient across the fault, which is almost invisible with the current scale.

5. How is the Hussain et al. (2016) 1-D screw dislocation model affected by the introduction of a dipping fault instead of a vertical fault? Is it negligible so that it is justified to use a model that assumes a vertical fault even when the actual fault it is dipping? Is it negligible because of the assumption that the fault has no dip-slip?

6. How are the GNSS velocities used in the modeling? It is not clear to me what is the contribution of this data set to the results. Were they jointly inverted with the InSAR data? If so, how were they weighted?

7. What is the shallow surface creep depth ($d_2$) found for the maximum likelihood solution (lines 109-110)?

8. From a quick look at the data only, it is very hard to tell if the velocity gradient is only due to fault locking or if there is a creep component in the motion (Figure 5). Do models assuming shallow creep perform better than those that assume no creep? Was this tested? If not, is there any other evidence in favor of shallow creep that justifies ignoring a case with no creep? Does the peak at 0 km for creep depth in Figure 6 (lines 199-200) favors the consideration of models with no shallow creep at all?

9. How is the result of a shallow surface creep rate of 2.2 mm/yr accounted for in the estimate of the moment deficit on this fault? It appears as it is ignored and the estimated slip rate is imposed to the whole fault. Wouldn't this overestimate the moment deficit?

10. Another possible overestimation of the moment deficit can be attributed to the locking depth, which is assumed to be at the bottom of the estimated seismogeinc zone, based on microseismicity relocation, but is shallower based on the modeling on this work. This is more or less discussed in section 6.1 of the manuscript, but I wonder if estimates of end-member moment deficits associated with the relevant bounds of the estimated locking depth (and accounting for shallow creep, if possible), could help bracket the range of earthquake magnitudes that could be expected at the fault, and explore those scenarios as well. Would this be possible and would it provide useful information to improve the discussion?

11. Could the discrepancy between the geological and geodetic slip rates be due to temporal variations of the slip rate (lines 182-184)?

12. How is the potential magnitude of a combined Padalarang-Lembang rupture estimated? Were the assumed slip rates, seismogenic zone depths, and return periods the same for both faults (those estimated for the Lembang fault in this work)?

**Minor corrections:**

13. Line 18: "recent history" might be more accurate that "recent memory".

14. Line 25: Is the Lembang basing the same as the Bandung basin? If not, please consider locating it on the map in Figure 1.

15. Line 79: "existing GNSS stations", is it station?

16. Lines 90-91: "The velocity for each GNSS station at mm level." Incomplete sentence, it probably was supposed to be part of the sentence before it.
17. Line 94: "5 km either…" Is it "5 km on either…"
18. Figure 7: the units of the Vs30 values are missing. Consider changing the label vs30_clipped to vs30.
19. Line 145: Figure 9 is mentioned in the main text before any mentions of Figure 8. Consider reordering the figures, although I perfectly understand why Figure 9 appears later.
20. Line 169: Figure 9 should be referenced here.
21. Line 211: "to jump larger across larger segment…", change to "to jump across larger segment…"
22. Line 215: "if the faults are late in its earthquake cycle", change to "if the faults are late in their earthquake cycle".
23. Line 245: "Sentinel-1¨ appears twice.

---

## Author Comment (AC1)

**RC1: 'Comment on egusphere-2022-1472', Anonymous Referee #1, 20 Apr 2023**

| No | Review | Response |
|---|---|---|
| 0 | This study investigates the slip rate of the Lembang fault, Indonesia, from Sentinel-1 images and GNSS measurements. The authors argue potential seismic hazards around the fault based on the estimated slip rate and previous recurrence of large earthquakes.

Because investigating seismic hazards posed by the Lembang fault is not only interesting from a scientific point of view but also important from a practical point of view, this study merits publication. However, the manuscript improves by addressing the following points. | We thank the reviewer for the constructive comments and suggestions. We have responded to each point below. |
| 1 | The authors' definition of locking (d1) and creep depths (d2) needs to be clarified. The equation below Line 100 indicates that the fault creeps from the surface to the d2 with a rate C, locks between d2 and d1, and slips below d1 with a rate S. If that is the case, d1 must be greater than d2. However, Figure 6 indicates that d2 can be greater than d1. To resolve my confusion, the authors must clearly define d1 and d2 with a figure, if necessary. | We will include a new figure (shown at the end of this review) to show the d1, d2 and related information. In our analysis we use a similar equation as Hussain et al (2016). |
| 2 | Figure 5 shows no discontinuities in the observed velocity field, indicating that surface creep is unlikely. Nonetheless, the authors assume the surface creeps on the fault. Is there any evidence for surface creeps from surface measurements, for example? | The evidence for shallow creep is from the GNSS stations located approximately ~50m either side of the fault. We agree that the discontinuity is not as obvious in the InSAR velocities. We attribute this to the significant noise present in the InSAR. This is seen clearly by the large spread in the grey points in Figure 5. |
| 3 | The Lembang fault dips to the north by 75 degrees (Line 24), but the modeling assumes a vertical fault (Line 100). Does this discrepancy affect the modeling much? | We agree that a more realistic model of the fault would take into account the dip. However, at 75 degrees, the fault is near vertical and so a simple vertical screw dislocation model captures the main aspects of the fault behavior we |

| | | |
|---|---|---|
| | | are looking for, namely the fault slip rate. The dip would only add a slight asymmetry to the profile weighted to the north. |
| 4 | The authors should say something about the subsidence of >50 mm/yr to the south of the Lambang fault because it is more visible than the displacement by faulting. | While this is not the main topic of the paper, we will add a short narrative on the subsidence in the main text. |
| 5 | The scenario seismic hazard delineated in Section 5 depends on the obtained fault-slip parameters, which have a fair amount of uncertainties. Then how does this scenario change with different fault-slip parameters within uncertainty bounds? | A difference in slip rate between 3.3–6.3 mm/yr results in a moment magnitude difference of 0.1. However, the return period difference between 170–670 years results in a moment magnitude range of 0.4.

Therefore most of the uncertainty in our models is in the return period. We have added text to the Discussion section of the manuscript explaining this. |
| 6 | I cannot understand how to look at Fig. 6. I understand that the black part at the center of a contour is where the probability is high. Then, how about the black part at the edge of and outside of the contour? | The points are the results from the full MCMC simulations. The black dots are the results from all our monte carlo simulations. The contours show the densest regions of the plot covering 86% of the data points. We have clarified this in the figure caption. |
| 7 | Related to the above comment, does Fig. 6 show that there are trade-offs between slip rate and creep depth, for example? | Yes, there are. This is a well known phenomenon in screen dislocation models where the slip rate trade-ffs against the locking depth. Hussain et al 2016 found no trade offs with the creep rate/depth. |
| 8 | Line 103: What is emcee? | Emcee is a Bayesian Markov Chain Monte Carlo algorithm developed by Foreman-Mackey et al. (2013) based on the work of Goodman and Weare (2010). We have made this clearer in the main text. |

[Figure]

Figure: Model setup

---

## Author Comment (AC2)

**RC2: 'Comment on egusphere-2022-1472', Anonymous Referee #2, 30 May 2023**

*Review of The seismic hazard from the Lembang Fault, Indonesia, derived from InSAR and GNSS data by E. Hussain et al. (2023)*

| General Comments | |
|---|---|
| In this study, the authors estimate the slip and shallow creep rates of the Lembang fault, Indonesia, by means of radar interferometry and continuous GNSS time series analysis. The results are used to make a first order assessment of the seismic hazard posed by this fault to the population living in neighbouring areas. This assessment is based on the superposition of the size of the population distribution and the ground accelerations predicted by the Global Earthquake Model OpenQuake engine for two maximum magnitude earthquakes (full rupture of the fault), corresponding to the moment deficit accumulation derived from the resulting slip rate and the upper and lower limits of the return period of large earthquakes on the fault. This is, thus, a study of a poorly understood fault which might have been overlooked because it hasn't had any recorded historical seismic activity. However, previous paleoseismology studies indicate large crustal earthquakes have been produced at this fault, with return periods longer than a century, and the present work confirms strain accumulation is occurring at the fault at rates that make it capable of producing shallow earthquakes with magnitudes larger than 6.5, at a fault embedded in a densely populated area. These are important results, not only to better understand the fault behaviour and to contribute to hazard assessment, but also to inform urban planning and risk for the nearby populations. The manuscript is very well written and has very nice figures that illustrate and support the main text. It is my opinion that the manuscript is almost ready for publication. However, I noted a few places where the discussion could be refined in order to improve its clarity. I list these points in the comments below. | We thank the reviewer for the constructive comments and suggestions. We have responded to each point below. |

| | Main comments: | |
|---|---|---|
| 1 | In the beginning of the manuscript, the authors state that "The fault dips to the north by about 75 degrees" (line 24). Why is a "constant fault dip of 60º" (line 125) used to estimate the area of the fault and the ground motion fields (line 132)? Also, when estimating the ground motion fields, the fault is modeled as two fault planes that "dip at 60º to the south" (line 132). Why the different orientation? | This is a mistake, we thank the reviewer for pointing this out. The fault dip should be 75 degrees to the north not 60. We will correct this error. |
| 2 | In the manuscript it is assumed that the motion at the fault plane is exclusively left-lateral strike-slip. Is there no evidence at all of dip-slip motion? Do the SAR and GNSS observations confirm this? | There is currently little evidence of dip slip motion across the fault from the geological offsets (Daryono paper). Our GNSS velocities do not show any normal component of motion with respect to the fault. In the InSAR this is difficult to test as the fault is oriented east-west and InSAR is insensitive to the north-south motion. |
| 3 | In figures 3a, b, c, the strong signal attributed to subsidence due to groundwater extraction obscures the signal due to interseismic deformation, which is the target of this study. Perhaps a change in the limits of the color scale could help enhance (at least in Figure 3c) the velocity gradient at the fault (if any), in order to confirm the assumption of no dip-slip rate accumulation. This could be done on a separate panel or an inset showing only the area of interest. A figure with profile A-A' showing a negligible vertical gradient could also help. | The vertical is dominated by subsidence. Which is the case in Bandung but also in Lembang.

The profile through the vertical velocities is given at the end of this document. There is no clear offset across the fault. The main signal is dominated by the long wavelength subsidence signal across the basin. The zero region is the spatial reference for the velocity maps. |
| 4 | Figure 3d could also benefit from a change in the color scale to enhance the horizontal velocity gradient across the fault, which is almost invisible with the current scale. | We have changed the colour scale on Figure 3d to better show the velocity difference across the fault. |
| 5 | How is the Hussain et al. (2016) 1-D screw dislocation model affected by the introduction of a dipping fault instead of a vertical fault? Is it negligible so that it is justified to use a model that assumes a vertical fault | We agree that a more realistic model of the fault would take into account the dip. However, at 75 degrees, |

| | | even when the actual fault it is dipping? Is it negligible because of the assumption that the fault has no dip-slip? | the fault is near vertical and so a simple vertical screw dislocation model captures the main aspects of the fault behavior we are looking for, namely the fault slip rate. The dip would only add a slight asymmetry to the profile weighted to the north. The screw dislocation model only solves for the fault-parallel component of motion so dip-slip motion is not accounted for. |
|---|---|---|---|
| 6 | | How are the GNSS velocities used in the modeling? It is not clear to me what is the contribution of this data set to the results. Were they jointly inverted with the InSAR data? If so, how were they weighted? | Yes, the GNSS and InSAR were jointly inverted in the model. And weighted by the uncertainty |
| 7 | | What is the shallow surface creep depth (d2) found for the maximum likelihood solution (lines 109-110)? | The maximum Likelihood solution for the creeped depth is: 0.1 km with a 90% confidence interval of 0--9.0 km. However, as we note in the Discussion the upper bound of this range is relatively unconstrained. |
| 8 | | From a quick look at the data only, it is very hard to tell if the velocity gradient is only due to fault locking or if there is a creep component in the motion (Figure 5). Do models assuming shallow creep perform better than those that assume no creep? Was this tested? If not, is there any other evidence in favor of shallow creep that justifies ignoring a case with no creep? Does the peak at 0 km for creep depth in Figure 6 (lines 199-200) favors the consideration of models with no shallow creep at all? | The evidence for shallow creep is from the GNSS stations located approximately ~50m either side of the fault. We agree that the discontinuity is not as obvious in the InSAR velocities. We attribute this to the significant noise present in the InSAR. This is seen clearly by the large spread in the grey points in Figure 5. A model with no creep would fit the InSAR data but not the GNSS. |
| 9 | | How is the result of a shallow surface creep rate of 2.2 mm/yr accounted for in the estimate of the moment deficit on this fault? It appears as it is ignored and the estimated slip rate is imposed to the whole fault. Wouldn't this overestimate the moment deficit? | Thank you for pointing this out. We originally did not account for the shallow creep as it is only equivalent to a release ~0.3% of the |

| | | |
|---|---|---|
| | | accumulated moment. However, in our new simulations we account for this release, which does not change the magnitude of the expected earthquake. |
| 10 | Another possible overestimation of the moment deficit can be attributed to the locking depth, which is assumed to be at the bottom of the estimated seismogeinc zone, based on microseismicity relocation, but is shallower based on the modeling on this work. This is more or less discussed in section 6.1 of the manuscript, but I wonder if estimates of end-member moment deficits associated with the relevant bounds of the estimated locking depth (and accounting for shallow creep, if possible), could help bracket the range of earthquake magnitudes that could be expected at the fault, and explore those scenarios as well. Would this be possible and would it provide useful information to improve the discussion? | The maximum likelihood estimate for the creep depth is very shallow (~0.1km), so the impact on the moment deficit is insignificant (equivalent to a release ~0.3% of the accumulated moment).

An (unlikely) locking depth of 3km, at the maximum likelihood slip rate (4.7 mm/yr) and accounted for shallow creep results in a moment magnitude deficit equivalent to a magnitude 6.2 earthquake.

And for a locking depth of 14.2km this results in a magnitude 7.0 earthquake.

We will include this range in the Discussion section of the manuscript. |
| 11 | Could the discrepancy between the geological and geodetic slip rates be due to temporal variations of the slip rate (lines 182-184)? | Yes, that is possible. This has been shown to be the case in Italy for example (Goodall et al., 2021). However, we don't have enough temporally constrained data to show this is the case for the Lembang Fault. |
| 12 | How is the potential magnitude of a combined Padalarang-Lembang rupture estimated? Were the assumed slip rates, seismogenic zone depths, and return periods the same for both faults (those estimated for the Lembang fault in this work)? | We converted the magnitudes to moment, summed the moment for the joint rupture and converted the total moment back to moment magnitude. |
| Minor Comments | | |

| 13 | Line 18: "recent history" might be more accurate that "recent memory". | Changed as suggested |
|----|---|---|
| 14 | Line 25: Is the Lembang basing the same as the Bandung basin? If not, please consider locating it on the map in Figure 1 | We have added this label to the map in Figure 1 |
| 15 | Line 79: "existing GNSS stations", is it station? | You are correct. It should be 'station' |
| 16 | Lines 90-91: "The velocity for each GNSS station at mm level." Incomplete sentence, it probably was supposed to be part of the sentence before it. | We rewrite this sentence |
| 17 | Line 94: "5 km either…" Is it "5 km on either…" | Edited as suggested |
| 18 | Figure 7: the units of the Vs30 values are missing. Consider changing the label vs30_clipped to vs30. | Edited and corrected |
| 19 | Line 145: Figure 9 is mentioned in the main text before any mentions of Figure 8. Consider reordering the figures, although I perfectly understand why Figure 9 appears later. | We have removed the earlier mention of Figure 9 |
| 20 | Line 169: Figure 9 should be referenced here. | Added as suggested |
| 21 | Line 211: "to jump larger across larger segment…", change to "to jump across larger segment…" | Corrected as suggested |
| 22 | Line 215: "if the faults are late in its earthquake cycle", change to "if the faults are late in their earthquake cycle" | Corrected as suggested |
| 23 | Line 245: "Sentinel-1¨ appears twice. | Removed duplicate |

[Figure]

Figure: Vertical velocities along profile.

---

## Author Response (AR2)

We have updated the Figure caption to Figure 10 to include the data citation as requested by the editorial office.